# Social Engagement and Elderly Health in China: Evidence from the China Health and Retirement Longitudinal Survey (CHARLS)

**DOI:** 10.3390/ijerph16020278

**Published:** 2019-01-18

**Authors:** Jin Liu, Scott Rozelle, Qing Xu, Ning Yu, Tianshu Zhou

**Affiliations:** 1Institute of Finance and Economics, Shanghai University of Finance and Economics, NO.777, Guoding Road, Yangpu District, Shanghai 200433, China; liu.jin@mail.shufe.edu.cn (J.L.); 15851732066@163.com (T.Z.); 2Research Institute for Agriculture, Farmer and Rural Society in China, Shanghai University of Finance and Economics, NO.777, Guoding Road, Yangpu District, Shanghai 200433, China; 3Rural Education Action Program, Freeman Spogli Institute for International Studies, Stanford University, Stanford, CA 94305, USA; rozelle@stanford.edu (S.R.); ningyu@stanford.edu (N.Y.)

**Keywords:** social engagement, elderly health, healthy aging, loneliness, 2SRI

## Abstract

This study examines the impact of social engagement on elderly health in China. A two-stage residual inclusion (2SRI) regression approach was used to examine the causal relationship. Our dataset comprises 9253 people aged 60 or above from the China Health and Retirement Longitudinal Survey (CHARLS) conducted in 2011 and 2013. Social engagement significantly improved the self-rated health of the elderly and reduced mental distress, but had no effect on chronic disease status. Compared with the rural areas, social engagement played a more important role in promoting the elderly health status in urban areas. Social engagement could affect the health status of the elderly through health behavior change and access to health resources. To improve the health of the elderly in China and promote healthy aging, the government should not only improve access to effective medical care but also encourage greater social engagement of the elderly.

## 1. Introduction

China has the largest elderly population in the world, with a projection of over 345 million people aged 60 years or above by the end of 2030, and this number is projected to increase rapidly over the next couple of decades [1,2]. Health is a critical determinant of healthy aging. Unfortunately, much is of concern in regard to the health of China’s elderly. On October 31, 2016, Wang Peian, the former deputy director of The Nation Health Commission of the People’s Republic of China, issued that among those aged 60 or above, 150 million had at least one chronic disease; approximately 7 million were diagnosed with dementia; 10 million, with permanent disability; and 40 million, with partial disability, and over 30% suffered from a variety of mental diseases. Identifying the factors that affect elderly health is a crucial first step in providing help.

Among the social determinants of health, social capital emerges as an important concept [3,4]. From a theoretical perspective, the association between social capital and health is well documented, and the positive relationship among social capital and better health outcomes has been examined in a number of countries and areas, including China [5,6,7], the United States [8], Japan [9,10], Europe [11,12], and Sub-Saharan Africa [13]. In addition, there is increasing evidence of the importance of social capital for promoting health among the aged [14].

Further, in old age, a large proportion of people live alone and have small social networks and low participation in social activities [15], making them more susceptible to feelings of loneliness. Loneliness is a common, painful, emotional experience, and it is a significant public health issue, especially among the elderly [16]. Increasing evidence has documented that loneliness in old age appears to be an important risk factor of being inactive [17] and worse health, including morbidity and mortality [18,19,20], depression [21,22], lower levels of self-rated physical health [15], and hypertension [23] as well as cardiovascular disease, diabetes, and migraine [24]. Using the Chinese Longitudinal Health Longevity Survey’s (CLHLS) last four waves of data from 2002 to 2011, another study found that loneliness has an adverse impact on cognitive functioning and vice versa [25].

There is an ongoing debate about how to measure social capital, and it is often unclear which activities form various aspects of it [26]. Moreover, there is disagreement about how to define social capital, whose definitions include such concepts as trust, norms, networks, relationship, civic participation, and social support, with social engagement as not included in these definitions. According to previous studies [27,28,29,30], three components of social capital have been assessed in the literature in relation to health outcomes: social participation, social networks, and social support. Social engagement, a real-life activity that results from association with one’s social ties, plays an important role in reinforcing social relationships, social support or social integration, resulting in maintaining better health and health outcomes [31,32] and is regarded as an important component of successful aging [33].

In later life, most adults no longer work or work fewer hours and have more time for social engagement [25,34]. Social engagement has become an important tool for older adults to obtain social resources. At present, there is global consensus on the importance of social engagement to active, healthy aging [35]. Social engagement not only helps to maintain older adults’ social activities and relations but also has a positive impact on their physical and mental health through communication and social support [36]. Therefore, it is imperative to investigate the influence of social engagement on elderly health. 

Previous research has established an association between social engagement and health promotion effect among old people. Some studies have suggested that social engagement can significantly improve an individual’s health. Using the data from the China Health and Retirement Longitudinal Study (CHARLS) conducted in the Zhejiang and Gansu Provinces in 2008, a previous study found that social engagement can significantly improve the self-rated health of residents 45 years and older [6]. Another studies replicated this result, using the data from the CHARLS conducted in 2011 [37]. Additionally, social capital could significantly reduce older adults’ emotional stress in rural areas [38]. 

Using the CLHLS, some studies aimed to gain more insight into the relationship between the elderly’s social engagement and self-rated health. Hu et al. noted that social engagement not only had a positive impact on elderly health but also played a positive role in disability prevention [39]. Lu et al. [35] found a causal, beneficial impact of social engagement on self-rated health among old people and that the impact of self-rated health on social engagement may be greater than that of social engagement on self-rated health. In addition, according to the Wisconsin study, the relationships between social participation and health vary by the type of trust and activity [8]. Other studies, however, have suggested that social activities do not affect health. For example, a study on rural residents of the Shandong province found that the impact of structural social capital, such as activity participation and social network, was not significant [40]. Meng and Chen [41] investigated the impact of social capital on self-rated health status, using the Chinese General Social Survey (CGSS) conducted in 2005, and found that social participation did not significantly affect residents’ self-rated health status. Similar conclusions were drawn by Snelgrove et al. [42] and d’Hombres et al. [43].

Across a range of studies, researchers have consistently shown the effect of social engagement on elderly health. Nonetheless, there are three major reasons why our investigation makes important contributions to the literature. 

First, this study builds upon the broader literature that links elderly health to social capital and focuses on the impact of social engagement on health. Most research on social engagement and promoting health focuses on only one indicator of social capital, which is an umbrella concept; thus, it is difficult to identify its aggregate impact. On the based of Putnam’s (2000) social capital theory [4], social capital consists largely of social networks, trust, and norms through social engagement. People who have more social engagement may create more social links or relationship, have more trust in others, and, thus, can be helpful in terms of sharing health information and knowledge, reinforcing access to quality health care and promoting individual health [13,44]. In addition, social engagement is useful for becoming integrated into their neighborhood network and forming a neighborhood-level understanding of the availability of and access to resources. This could be helpful for the elderly to obtain material and emotional support of their neighbors and, thus, support their health [28,45]. Further, participating in social activities is a valuable approach to improving quality of life and reducing the burden associated with declining health and functioning as older adults age [46]. 

Second, this study recognizes that there is no “typical” elderly, as the health and functional status of the elderly are diverse. Thus, it is often difficult to measure and compare their health status. Despite growing evidence that suggests a relationship between social engagement and elderly self-report health, more objective measures are needed. The elderly often suffers from a variety of serious health-related consequences, such as decreased economic resources and increased medical burden to themselves and their families. These could further affect the elderly’s psychological well-being and health behavior [47]. A pioneering study found that these serious health-related consequences may even raise the rate of suicide in elderly people, especially the rate for the elderly in rural areas [48]. What’s important, based on the “China’s Medium-to-Long Term Plan for the Prevention and Treatment of Chronic Diseases (2017–2025)”, which was issued by the General Office of the State Council of the People’s Republic of China on January 22, 2017, there are more than 260 million people who suffer from chronic diseases in China and the rate of death that results from chronic diseases is 86.6%, when disease burden accounts more than 70% for all disease burdens. In the reference literature, it has been provided that neurodegenerative and neuropsychiatric conditions that can affect older people, such as dementia and Parkinson’s disease, are often characterized by significant psychological deficits that reduce their overall quality of life [49].

To measure the influence of social engagement on the health and well-being of the elderly, we use objective and subjective approaches to measure health, including self-rated health, degree of psychological distress and the numbers of chronic diseases.

Third, a lack of sufficient sample sizes/data in the past may have caused statistically biased estimates [37]. The current study analyzed the data derived from the 2011 and 2013 waves of the CHARLS and estimates how social engagement affects elderly health in a more comprehensive way, providing evidence on healthy aging in China.

This paper proceeds as follows. Section 2 presents the hypotheses, model specification, data source, and variable selection. Section 3 provides the results. Section 4 presents potential mechanisms for the findings, and Section 5 concludes the paper. 

## 2. Hypotheses, Methods, Data, and Variables

### 2.1. Hypotheses

On the basis of Pierre Bourdieu’s (1986) social capital theory [50], as the measure of social capital, social engagement reflects the individual’s social network, social participation, and social support, which not only are a kind of social capital but also reflect the individual’s ability to acquire social network resources. More details are provided in the Figure 1. Based on this, the following hypotheses were developed:

**Hypothesis** **1.**
*Social engagement improves the health of older adults.*


**Hypothesis** **2.**
*Social engagement has positive effects on elderly health through two pathways: (1) health behaviors of the elderly (such as whether to exercise), which is associated with health among older people; and (2) access to health care, which exerts an indirect effect on the relationship between social engagement and elderly health.*


### 2.2. Methods

We adopt the following general framework of empirics:(1)$$H_{i}=\alpha_{0}+\alpha_{1}Activity_{i}+\delta X_{i}+\varepsilon_{i}$$

In Equation (1), i refers to the individuals in the survey, $H_{i}$ denotes the health status of the individual, and $Activity_{i}$ represents whether the individual participate in social activities; $X_{i}$ refers to other control variables, $\varepsilon_{i}$ is the error term, and $\alpha_{1}$ is the parameters to be estimated, reflecting the impact of social engagement on health.

The model, however, may have endogeneity problems, which can result in biased results. There are two reasons for such endogeneity. First, social engagement depends on unobservable preferences (such as the confidence in expectations), which may lead to self-selection or missing variables. Second, there may be a reverse causality between social engagement and health, which means that health status could play a role in older adults’ social engagement.

We turn to the instrumental-variable method with two-stage regression estimation to address these two problems. Our instrumental variables are: (1) whether the villages/ communities have chess and card rooms or activity centers for the elderly and (2) how many bus lines reach the village/community. The two variables are believed to affect the elderly’s social engagement but are irrelevant to health status, belonging to exogenous variables. In addition, the number of instrumental variables is greater than the number of endogenous variables, and an over-identification test can be performed to identify its validity.

From the perspective of the health economics literature, dependent variables are often nonlinear forms, such as restricted dependent variables, counting variables, or a skewed distribution. Two-stage residual inclusion (2SRI) is a consistent and effective estimation method to solve the endogeneity of nonlinear models [51,52]. Model (1) can be presented as follows:(2)$$\mathrm{Stage}\;1:\;SA_{i}=\beta_{0}+\beta_{1}IV_{i}^{1}+\beta_{2}IV_{i}^{2}+\varphi X_{i}+\mu_{i}$$
(3)$$\mathrm{Stage}\;2:\;H_{i}=M(\alpha_{0}+\alpha_{1}\bar{SA_{i}}+\delta X_{i}+\gamma {\overset{\wedge }{x}}_{\mu i})+e^{2SRI}$$

In Equation (3), ${\overset{\wedge }{x}}_{\mu i}$ is the residual fitting value of the first-stage regression equation calculation.

### 2.3. Data

The primary database used in this work is drawn from the China Health and Retirement Longitudinal Study (CHARLS). CHARLS has received critical support from Peking University, the National Natural Science Foundation of China, the Behavioral and Social Research Division of the National Institute on Aging and the World Bank. CHARLS is a nationally representative longitudinal survey of persons in China 45 years of age or older and their spouses, including assessments of social, economic, and health circumstances of community residents [53]. All data will be made public one year after the end of data collection. CHARLS adopts multi-stage stratified PPS sampling. As an innovation of CHARLS, a software package (CHARLS-GIS, CHARLS, Peking University, Beijing City, China) is being created to make village sampling frames. The CHARLS questionnaire includes the following modules: demographics, family structure/transfer, health status and functioning, biomarkers, health care and insurance, work, retirement and pension, income and consumption, assets (individual and household), and community level information. 

The baseline national wave of CHARLS is being fielded in 2011 and includes about 10,000 households and 17,500 individuals in 150 counties/districts and 450 villages/resident committees (or villages) from 28 provinces. Furthermore, the CHARLS respondents are followed up every two years, using a face-to-face computer-assisted personal interview.

Since we are interested in exploring the relationship between the health of the elderly and the social engagement, we restrict our attention to the subsample of the elderly in China, and further limit our sample to respondents who are aged 60 or above. By eliminating the missing values, the final sample contains 9253 individuals, with 5626 from rural areas and 3627 from urban areas. 

### 2.4. Variables

#### 2.4.1. Health

Since health is a multi-dimensional and general concept, we examine three health indicators: self-rated health, mental health, and chronic diseases. Self-rated health is a common indicator of health status. In CHARLS, self-rated health is obtained by asking participants, “What do you think of your health? Is it excellent, very good, good, poor, or bad?” For regression model applicability, we redefine “excellent,” “very good,” and “good” as good health and assign a value of 1; we redefine “poor” and “bad” as bad health, and assign a value of 0.

One of the basic principles of category merging is to combine similar categories together and to maximize differences between categories in terms of health and physical function, and there is a significant relationship between depression symptoms. This study calculated the scores of different self-rated health categories in terms of mental health and the number of chronic diseases. More details about self-rated health are provided in the Table 1. It is found that the difference between their health as “excellent” or “very good” and “good” is smaller than that of those health as “good” and “poor” or “bad”, so “excellent”, “very good”, and “good” are combined. 

We further use the degree of psychological distress to measure mental health. In CHARLS, the degree of individual psychological distress is obtained by asking, “How did you feel and behave last week?” After receiving the response, the participant is asked, “Was it a little or never, not much, sometimes, half of the time, or most of the time?” There were 10 questions in total, and four answers were assigned values of 0–3, with a total score of 0–30; the higher the score, the higher the degree of psychological distress.

Finally, to address potential bias related to misreporting, the number of chronic diseases was used as an indicator of health. For the CHARLS, information about chronic diseases is obtained by asking participants, “Have you been told by doctors or do you know that you have at least one of the following 14 diseases: hypertension, dyslipidemia, diabetes/high blood, cancer or other malignant tumors, chronic lung disease, liver disease, heart disease, stroke, kidney disease, stomach disease/digestive system diseases, emotional and spiritual diseases, memories related diseases, arthritis/rheumatism, or asthma?” A “yes” answer is assigned a value of 1; otherwise, the value is 0. Finally, the total number of chronic diseases is obtained.

#### 2.4.2. Social Engagement

The key explanatory variable in this study is social engagement. In CHARLS, information about social engagement is obtained mainly through the question, “Have you participated in the following social activity in the past month?” for which there are 12 options (interacted with friend; play ma-jong, play chess, play cards, or went to community club; provided help to family, friends, or neighbors who do not live with you and did not pay you for the help; went to a sport, social, or other kind of club; took part in a community-related organization; done voluntary or charity work; cared for a sick or disabled adult who does not live with you and who did not pay you for the help; attended an educational or training course; stock investment; used the Internet; other; none of these. What’s worthy, the interview cannot select” None of these” together with any other answer).

Since the two options of “participate in other activities” and “none of the above” are in conflict with the other 10 activities, it is impossible to select them simultaneously, and the remaining 10 options are selected for analysis. To facilitate the interpretation of the regression results, we reassigned the values of these 10 activities; that is, the value of participating in each activity is 1; otherwise, it is 0. The variable of “participates in social activities” is the total for the 10 activities.

#### 2.4.3. Other Control Variables 

This study controls for individual, household, and village/community variables. The individual and household variables are gender, age, marital status, educational level, family income, whether one smokes, and whether one lives in an urban area. The following variables reflect the characteristics of villages/communities: the total permanent population of villages, per capital disposable income of villages, average price of new houses in villages, regional variables, and so forth.

## 3. Results

### 3.1. Descriptive Statistics

Table 2 presents the descriptive statistics for all variables, and the sample is divided into two groups: urban and rural. The results show that the percentage of people with good self-reported health was 68.6%, the mean degree of psychological distress was 10.5, and everyone in the sample suffered from at least one chronic disease. Almost half (49%) had access to social engagement in the past month. A little more than half were male (50.1%), and the mean age was 68.4 years. In addition, 37% smoked and 78.5% were married; 55% had less than a primary school education, 24.1% had primary school or private school education, and 20% completed a junior high school education or above. The mean annual household income of the sample was about 20,000 Yuan.

The self-rated health and degree of psychological distress of the elderly in rural areas were poorer than that of their urban peers, but the number of rural people with chronic diseases was less than that of their urban peers. Of older adults in urban areas, 54.8% participated in social activities, nearly 10% higher than for their rural counterparts. In urban areas, the numbers of village/community chess and card rooms or elderly activity centers, bus lines, and urban communities were also higher than those of rural villages. In addition, the educational level and family economic status of urban elderly people were relatively high.

In regard to village characteristics, there was a difference between cities and villages in terms of disposable income per capita, permanent population, and housing prices. For example, the average disposable income per capita of all villages in the sample was close to 5000 Yuan, but the per capita disposable income of rural villages was only about 3700 Yuan, which is 6800 Yuan less than in urban areas, where the average disposable income is 10,500 Yuan. In terms of geographic location, the proportion of the elderly in the western region was the highest (36.8%), followed by the eastern (29.9%) and central regions (28.5%), with the northeast region’s having the lowest proportion (4.8%).

### 3.2. Influence of Social Engagement on Elderly Health

Table 3 reports the estimation results of the 2SRI. The second column of the table shows the results for the application of the first-stage instrumental variable method to solve the endogeneity problem. The results showed that the instrumental variables had a significant positive effect on the endogenous variables. Specifically, the more elderly adults had access to chess and card rooms or elderly activity centers as well as bus lines in villages/communities, the higher their participation in social activities.

Table 3 also presents the estimates of the effect of social engagement on three indicators of health status. 

#### 3.2.1. Self-Rated Health

In terms of self-rated health, the marginal effect value of social engagement remains significant and positive (0.1293 with a *p*-value of 0.01) when other variables were controlled. This suggests that social engagement is an important channel for the elderly to have access to health care information, which can be used to improve their own health. 

There were urban-rural and regional differences in regard to self-rated health status, which resulted from living standards. The relationship between age and self-rated health is non-linear: the marginal effect value of age remains significant and negative (−0.0619 with a *p*-value of 0.01), but the value of age square is significant and positive (0.0004 with a *p*-value of 0.01). The marginal effect value of the gender dummy is significant and positive. This result confirms that male elderly health is better than their female peers. The self-rated health of elderly with spouses is better than those without spouses. Education and family economic status has no significant impact on the elderly’s self-rated health.

However, the measure of self-rated health is dichotomized, a great deal of information could be lost if those who classify their health as excellent are grouped with those who say good (we thank an anonymous reviewer for this observation). The self-rated health status is a multiple choice and case-specific, a multinomial Logit model is used to examine the effect of social engagement on health among the elderly. More detail is presented in Table 4. The empirical result shows that after adjustment for other variables, based on good self-rated health group, significant associations (*p* < 0.001) are found between social engagement and poor and very bad self-rated health status, with more social engaged elderly reporting less poor or very bad self-rated health. This result confirms that social engagement could reduce the odds ratio of poor or very bad self-rated health of the elderly. The negative association is also demonstrated, but no significant.

#### 3.2.2. Degree of Psychological Distress

In Table 3, as expected, the empirical results also show that the social engagement has a significant and positive on degree of psychological distress of the elder (−5.5571 with a *p*-value of 0.01). This result confirms our expectation that active social engagement has been shown to be associated with better mental health among older people. 

With regard to other control variables, the relationship between age and degree of psychological distress is also non-linear; the marginal effect value of age remains significant and positive (0.7207 with a *p*-value of 0.01), while the value of age square is significant and negative (−0.0057 with a *p*-value of 0.01). The degree of psychological distress of men is significantly lower than that of women. Smoking helped these older adults to relieve their psychological pressure and distress. Compared with the elderly who had received less than a primary school education, the elderly who received more education has higher degree of psychological distress. The distress of people with a spouse was less than 1.3 points lower than those without a spouse, significant at *p* < 0.01.

In addition, the degree of psychological distress of the elderly is significantly different between urban and rural areas, with the degree of psychological distress of the elderly in rural areas as higher than that of their urban counterparts. The disposable income per capita of villages has a significant positive effect on degree of psychological distress; the higher the disposable income per capita of villages, the worse the mental health of the residents. One possible reason is that, in regions with a high level of economic development, labor market competition was more intense, which increased the psychological distress of local residents and made their mental health level decline. Housing prices also could significantly decrease the degree of psychological distress of residents, as the housing prices in the area were low, thus, not leading to economic and psychological pressure.

#### 3.2.3. Number of Chronic Diseases

Although the relation is negative between social engagement and the number of chronic diseases among the elderly, the result is not significant. At the same time, age has a significant and positive on the number of chronic diseases among the elderly. The number of older people with chronic diseases in urban areas is higher than that of their rural counterparts. The number of smokers who suffered from chronic diseases is significantly higher than that of non-smokers. In addition, from the perspective of village/community variables, the higher the local housing price, the significantly greater the number of chronic diseases suffered by residents.

### 3.3. Robustness Test

To test the robustness of these estimates, we used the variable transformation method in terms of the amount of social engagement. The corresponding regression results are presented in Table 5. As shown in the table, the coefficient signs and the significance of the main explanatory variables as well as the other control variables did not change much. Based on the above analysis, we believe that the results of the study were robust.

### 3.4. Effect of Social Engagement on Elderly Health: Urban-Rural Differences

Table 6 shows the effect of social engagement on elderly health between urban and rural areas. The results indicate that social engagement significantly influenced the self-rated health and degree of psychological distress of the urban and rural samples. Nevertheless, there are differences between urban and rural areas. Social engagement plays less on rural older people than urban peers. Since rural areas have relatively less developed infrastructure for elderly care services (such as outdoor fitness equipment, elderly activity centers, and elderly associations), the social networks among rural elderly people is relatively constant and narrow, and they have much less access to health prevention and medical information. The elderly in urban areas receive an income after retirement, which results in less financial pressure and more time for leisure activities and exercise. Nevertheless, older people in rural areas have to keep working until they were physically unable to do so, which could have resulted in pressure and would have left less time to engage in leisure activities and exercise.

This study also used the method of variable transformation to investigate the effect of the number of social engagement on health of urban and rural elderly individuals. As seen in Table 5, this major explanatory variable’s signs and significance changed little, which indicated that the results of this study are robust.

## 4. How Social Engagement Influences the Health of the Elderly

Although active social engagement has been shown to be associated with better health outcomes across our study and other literature to date, there is unclear or insufficient evidence that how social engagement leads to better health. New research concerning the relationship would be offered, while an increasing literature has considered the issue. In our opinions, it assumed that social engagement would be associated with elderly health through social engagement’s effect on health behaviors and access to medical and health resources. To verify these two mechanisms, we estimate the following equation:(4)$$Y_{i}=\varphi_{0}+\varphi_{1}\times SA_{i}+\gamma \times X_{i}+\varpi_{i}$$

The dependent variables are healthy behavior and access to medical and health resources. Healthy behavior is determined from whether they exercise regularly. On the CHARLS, respondents are asked, “Do you normally do intense, moderately intense, or leisure activities for at least 10 minutes a week?” (on the CHARLS, respondents were asked to think only about those physical activities that they did for at least 10 minutes at a time) the response to which is a binary variable. For access to medical and health resources, we used the question, “Did you use outpatient services in the past month?” the response to which also was a binary variable.

Based on Equation (4), the results are shown in Table 6. The results indicate that, in all samples, the effect of participation in social engagement is significant and positive, the probability of physical exercise and of choosing a healthy way of life, which is helpful to maintaining or improving health status. The results also show that, for those who participated in social engagement, the probability of seeing a doctor is significantly increased in the past month, compared with those who do not participate in social activities. As we expected, the results confirm social engagement could help to make older adults obtain medical and health information, which would be useful to promote elderly health.

Finally, we use the method of transforming variables for verification. The results are shown in Table 7. The results show that the effect of the explanatory variables is consistent with previous results.

## 5. Discussion

Population aging poses a significant challenge to individual and social health, how to facilitate healthy aging by promoting health and well-being among the elderly is a topic of the utmost importance. Employing detailed longitudinal data from the CHARLS conducted in 2011 and 2013, we applied a 2SRI approach to gain more insight into the association between social engagement and elderly health. We found that social engagement could improve the elderly’s self-rated health and mental health through changing their health behaviors and access to health care and resources and that the urban elderly’s health status in urban was better than that of their rural counterparts.

Our findings suggest policy interventions to maintain and improve health and well-being among the elderly. In addition to comprehensive public health actions, the government should help to create an elderly-friendly environment. It should build activity centers and increase public service fiscal expenditures. In addition, social resources should be mobilized for investment in the elderly-care industry. Finally, the government needs to increase the construction of rural pension service facilities to narrow the gap between urban and rural areas. 

This study enriches and expands on previous research and contributes to the literature in three ways. First, we used an innovative research perspective. We focused a community-supporting environment that meets the needs of the elderly and helps them to be actively engaged in their health. We also focused on the impact of interpersonal communication (i.e., social engagement) on the health of the elderly. Further, we measured both subjective and objective indicators of elderly health, providing a more accurate and comprehensive understanding. Second, we adopted new instrumental variables for causal 2SRI estimations: whether the village/community has a chess and card room or activity center for the elderly and how many bus lines be accessible this village/community. Third, our conclusions and policy recommendations are widely applicable to China and other countries with a similar elderly culture.

Though this work provides evidence for increasing social engagement on elderly health, we consider several potential limitations. 

First, if the elderly in rural area would migrate to the cities or towns, it would be necessary to gain more insight into the relationship of social engagement and health, and differentiate between the resources only accessible through individual connections and resources provided through social engagement.

Second, in social capital literature, neighborhood social capital or community-level social capital is important for elderly health and health behavior. But as most of the elderly in China, especially those in rural China, have permanent address, available scales could be lacking and fail to capture multiple dimensions of community-level social capital, suggesting the potential generalizability of our study to alternative models, such as hierarchical liner model and the nested level model (we thank an anonymous reviewer for this observation).

## 6. Conclusions

In actually, aging of society is a significant challenge to public health [54]. Thus, an elderly-friendly supportive environment and measures to promote healthy aging include providing opportunities for the elderly to participate in social engagement, improving the living environment, and promoting age diversity in the work environment. In order to improve the quality of the elderly life, it would be worthy of increasing the control of negative emotions, and self-efficacy by social engagement [49]. Our results here do indeed suggest that increasing social engagement has a positive impact on elderly health and is important for successful aging [33]. With China’s urbanization and deepening reform of the household registration system, the elderly in rural areas may have to adapt to new environments after they migrate to cities, where they may have scant social connections and. We believe whether a policy that aims to increase access to social engagement should target friendly facilities and health promotion. 

## Figures and Tables

**Figure 1 ijerph-16-00278-f001:**
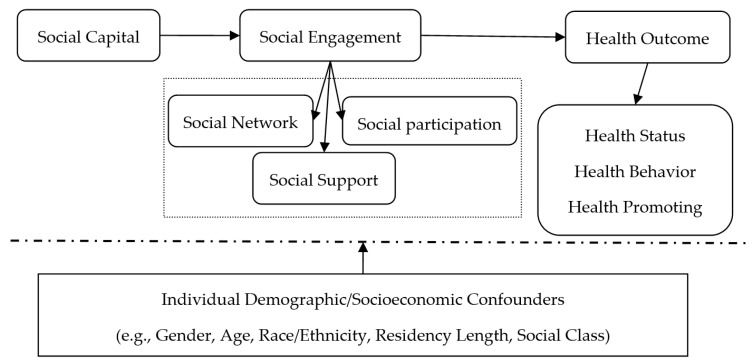
Conceptual model of social engagement processes on elderly health outcomes.

**Table 1 ijerph-16-00278-t001:** Different self-reported health statuses.

Classification Based on Self-reported Health	Obs	Mean	Std. Dev	Min	Max
Self-reported health is excellent:					
Individual’s degree of psychological distress	661	7.799	3.984	0	25
Individual’s number of chronic diseases	661	0.951	1.153	0	8
Self-reported health is very good:					
Individual’s degree of psychological distress	1217	8.680	4.024	1	27
Individual’s number of chronic diseases	1217	1.115	1.248	0	8
Self-reported health is good:					
Individual’s degree of psychological distress	4467	10.135	4.256	2	28
Individual’s number of chronic diseases	4467	1.548	1.370	0	8
Self-reported health is poor:					
Individual’s degree of psychological distress	2182	12.241	4.577	1	30
Individual’s number of chronic diseases	2182	2.086	1.574	0	9
Self-reported health is very bad:					
Individual’s degree of psychological distress	726	13.017	4.928	3	29
Individual’s number of chronic diseases	726	2.358	1.646	0	10

Obs: number of observations; Std. Dev.: Standard Deviation.

**Table 2 ijerph-16-00278-t002:** Sample characteristics.

Variable	Definition	All	Area of Residence	
			Rural	Urban
Health_self	Self-rated: good = 1, bad = 0	0.686 (0.464)	0.649 (0.477)	0.743 (0.437)
Degree of psychological distress	Score of psychological distress	10.500 (4.591)	11.037 (4.666)	9.667 (4.344)
Chronic	Number of chronic diseases	1.639 (1.471)	1.580 (1.427)	1.730 (1.532)
Social Engagement	Participate = 1, otherwise = 0	0.490 (0.500)	0.453 (0.498)	0.548 (0.498)
Age	Age	68.439 (7.135)	68.360 (7.037)	68.562 (7.283)
Gender	Male = 2, female = 0	0.501 (0.500)	0.506 (0.500)	0.494 (0.500)
Spouse	Married, cohabitating = 1, otherwise = 0	0.785 (0.411)	0.781 (0.413)	0.792 (0.406)
Illiterate	Below primary school = 1, otherwise = 0	0.559 (0.497)	0.645 (0.479)	0.424 (0.494)
Elementary	Private education or elementary education = 1, otherwise = 0	0.241 (0.428)	0.237 (0.425)	0.248 (0.432)
Secondary	Secondary school or above = 1, otherwise = 0	0.200 (0.400)	0.118 (0.323)	0.328 (0.469)
Smoking	Yes = 1, No = 0	0.369 (0.482)	0.385 (0.487)	0.343 (0.475)
Income	Annual household income(10,000 Yuan)	2.058 (4.237)	1.303 (2.841)	3.230 (5.569)
Per-income	Village disposable income per capita (10,000 Yuan)	0.494 (0.587)	0.372 (0.434)	0.683 (0.724)
Total-pop	Village permanent population (10,000)	0.326 (0.345)	0.190 (0.152)	0.535 (0.442)
Price-apart	Village average prices of new houses (Yuan/m^2^)	0.268 (0.767)	0.190 (0.825)	0.389 (0.651)
Urban	Urban areas (1 = yes, 0 = no)	0.392 (0.488)	--	1.000 (0.000)
East	Live in eastern area = 1, other = 0	0.299 (0.458)	0.286 (0.452)	0.321 (0.467)
Middle	Live in middle area = 1, other = 0	0.285 (0.452)	0.296 (0.456)	0.269 (0.444)
West	Live in western area = 1, other = 0	0.368 (0.482)	0.382 (0.486)	0.347 (0.476)
North_east	Live in north-east area = 1, other = 0	0.048 (0.211)	0.036 (0.191)	0.063 (0.243)
Acti_card	Chess and card room/elderly activity left = 1, otherwise = 0	0.456 (0.498)	0.289 (0.454)	0.714 (0.452)
Bus_line	Number of bus lines to village/community	2.221 (4.047)	1.210 (3.088)	3.790 (4.790)

Standard deviations are in parentheses.

**Table 3 ijerph-16-00278-t003:** 2SRI estimates of the effect of social engagement on the health of the elderly.

Variable	Social Engagement	Self-Rated Health	Degree of Psychological Distress	Number of Chronic Diseases
Social Engagement	--	0.1237 ** (0.0094)	−0.5571 ** (0.0951)	−0.0175 (0.0310)
Age	0.0731 ** (0.0128)	−0.0619 ** (0.0190)	0.7207 ** (0.1632)	0.1296 * (0.0548)
Agesqr	−0.0006 ** (0.0001)	0.0004 ** (0.0001)	−0.0057 ** (0.0012)	−0.0008 * (0.0004)
Gender	−0.0334 ** (0.0130)	0.0625 ** (0.0146)	−1.5524 ** (0.1369)	−0.1321 ** (0.0456)
Spouse	−0.0772 ** (0.0135)	0.0715 ** (0.0221)	−1.2090 ** (0.2052)	0.1279 * (0.0660)
Elementary	0.0506 ** (0.0129)	−0.0159 (0.0180)	0.1672 (0.1699)	0.0245 (0.0540)
Secondary	0.1517 ** (0.0154)	−0.0711 † (0.0415)	0.8319 * (0.3715)	−0.1174 (0.1189)
Smoking	0.0309 * (0.0128)	0.0021 (0.0140)	0.5490 ** (0.1296)	−0.1287 ** (0.0438)
Income	0.0096 * (0.0039)	−0.0022 (0.0021)	0.0297 † (0.0181)	−0.0087 † (0.0055)
Per-income	0.0332 ** (0.0098)	-0.0201 (0.0134)	0.2352 * (0.1138)	0.0013 (0.0387)
Total-pop	0.0206 (0.0179)	−0.0420 * (0.0178)	0.1568 (0.1629)	−0.0275 (0.0569)
Price-apart	−0.0072 (0.0070)	0.0042 (0.0066)	−0.1754 ** (0.0463)	0.0303 † (0.0181)
East	0.0099 (0.0130)	0.0671 ** (0.0120)	−1.0677 ** (0.1194)	−0.3854 ** (0.0386)
Middle	0.0378 ** (0.0128)	−0.0021 (0.0152)	−0.1200 (0.1443)	−0.1938 ** (0.0461v
North_east	0.0389 † (0.0255)	−0.0118 (0.0258)	−0.6629 ** (0.2426)	−0.0601 (0.0878)
Urban_nbs	0.0153 * (0.0134)	0.0483 ** (0.0135)	−0.6256 ** (0.1280)	0.0776 * (0.0425)
Acti_card	0.0239 ** (0.0116)	--	--	--
Bus_line	0.0024 * (0.0014)	--	--	--
Residual	--	0.7782 ** (0.2430)	−9.7327 ** (2.1478)	1.2514 † (0.6925)
Constant	--	--	−4.9815 (4.7079)	−3.8736 * (1.5976)
Wald chi^2^	418.30 **	487.94 **	--	--
Pseudo R^2^	0.0384	0.0451	--	--
F-value	--	--	45.71 **	13.44 **
R^2^	--	--	0.0724	0.0235

All models are marginal effects with robust standard errors in parentheses. † *p* < 0.10, * *p* < 0.05, ** *p* < 0.01.

**Table 4 ijerph-16-00278-t004:** Estimates of the effect of social engagement on the self-rated health of the elderly.

Variable	Excellent	Very Good	Poor	Very Bad
	Based on Good	Based on Good	Based on Good	Based on Good
Social Engagement_num	−0.0215 (0.0872)	−0.0521 (0.0671)	−0.421 *** (0.0545)	−1.1219 *** (0.0918)
Age	0.0327 (0.1423)	−0.2940 ** (0.1201)	0.2176 † (0.1135)	0.3299 † (0.2027)
Agesqr	−0.0003 (0.0011)	0.0022 ** (0.0009)	−0.0015 † (0.0009)	−0.0023 (0.0015)
Gender	0.4525 *** (0.1232)	0.2129 * (0.0951)	−0.2398 ** (0.0834)	−0.1343 (0.1529)
Spouse	−0.2671 † (0.1636)	0.1667 (0.1461)	−0.3185 * (0.1272)	−0.4003 (0.2578)
Elementary	−0.1476 (0.1439)	−0.2745 * (0.1186)	0.0274 (0.1023)	−0.0487 (0.1947)
Secondary	0.2317 (0.2979)	−0.3789 (0.2586)	0.2935 (0.2395)	0.2683 (0.4964)
Smoking	-0.0470 (0.1141)	-0.1405 (0.0926)	−0.0316 (0.0798)	−0.1176 (0.1466)
Income	0.0199 * (0.0085)	0.0003 (0.0128)	0.0156 (0.0114)	−0.0082 (0.0428)
Per-income	0.1010 (0.0895)	−0.0853 (0.0786)	0.0813 (0.0786)	0.0897 (0.1485)
Total-pop	0.0302 (0.1567)	0.1092 (0.1184)	0.2449 * (0.1024)	0.2011 (0.1587)
Price-apart	−0.0375 (0.0731)	0.0356 (0.0401)	−0.0166 (0.0359)	−0.0263 (0.0620)
East	0.6458 *** (0.1098)	0.2333 ** (0.0831)	−0.1435 * (0.0697)	−0.4913 *** (0.1132)
Middle	0.4052 ** (0.1267)	−0.1365 (0.1032)	0.0291 (0.0844)	−0.0279 (0.1529)
North_east	1.0334 *** (0.1851)	−0.3312 † (0.1891)	0.1436 (0.1418)	0.0455 (0.2373)
Urban_nbs	0.0862 (0.1082)	−0.0593 (0.0890)	−0.2781 *** (0.0787)	−0.1226 (0.1225)
Residual	−2.0769 (1.6646)	2.0108 (1.5314)	−3.3777 * (1.4023)	−4.2236 (3.0349)
Constant	−2.2598 (4.3087)	7.3443 * (3.5080)	−6.1968 (3.2801)	−10.6774 (5.4233)
Wald chi^2^	656.65 ***			
Pseudo R^2^	0.0292			
Log pseudolikelihood	−12102.341			
Number of obs	9253			

*Note.* All models are marginal effects with robust standard errors in parentheses. The first-stage result of 2SRI is not given. † *p* < 0.10, * *p* < 0.05, ** *p* < 0.01, *** *p* < 0.001.

**Table 5 ijerph-16-00278-t005:** Estimates of the effect of social engagement on the health of the elderly.

Variable	Self-rated Health	Degree of Psychological Distress	Number of Chronic Diseases
Social Engagement_num	0.0696 ** (0.0057)	−0.3776 ** (0.0476)	−0.0004 (0.0171)
Age	−0.0518 ** (0.0190)	0.6411 ** (0.1620)	0.1301 * (0.0551)
Agesqr	0.0004 ** (0.0001)	-0.0051 ** (0.0012)	−0.0008 * (0.0004)
Gender	0.0595 ** (0.0146)	−1.5308 ** (0.1365)	−0.1324 ** (0.0456)
Spouse	0.0644 * (0.222)	−1.1533 ** (0.2041)	0.1272 * (0.0661)
Elementary	−0.0169 (0.0180)	0.1615 (0.1694)	0.0251 (0.0540)
Secondary	−0.0725 † (0.0415)	0.8167 * (0.3692)	−0.1158 (0.1191)
Smoking	0.0045 (0.0140)	0.5240 ** (0.1293)	−0.1285 ** (0.0438)
Income	−0.0020 (0.0021)	0.0271 (0.0178)	−0.0087 (0.0055)
Per-income	−0.0172 (0.0134)	0.2102 † (0.1132)	0.0017 (0.0388)
Total-pop	−0.0454 * (0.0179)	0.1749 (0.1617)	−0.0271 (0.0569)
Price-apart	0.0025 (0.0065)	−0.1618 ** (0.0463)	0.0303 † (0.0181)
East	0.0703 ** (0.0120)	−1.0910 ** (0.1193)	−0.3853 ** (0.0387)
Middle	−0.0002 (0.0153)	−0.1416 (0.1438)	−0.1934 ** (0.0462)
North_east	−0.0081 (0.0259)	−0.6981 ** (0.2423)	−0.0598 (0.0879)
Urban_nbs	0.0475 ** (0.0135)	−0.6282 ** (0.1277)	0.0779 † (0.0426)
Residual	0.7144 ** (0.2437)	−8.9856 ** (2.1369)	1.2255 † (0.6962)
Constant	--	−2.6456 (4.6718)	−3.8861 * (1.6040)
Wald chi^2^	441.72 **	--	--
Pseudo R^2^	0.0449	--	--
F-value	--	47.70 **	13.44 **
R^2^	--	0.0748	0.0234

All models are marginal effects with robust standard errors in parentheses. The first-stage result of 2SRI is not given. † *p* < 0.10, * *p* < 0.05, ** *p* < 0.01.

**Table 6 ijerph-16-00278-t006:** Estimates of the effect of social engagement by urban versus rural.

Variable	Self-Rated Health		Degree of Psychological Distress		Number of Chronic Diseases	
	Urban	Rural	Urban	Rural	Urban	Rural
Part I						
Social Engagement	0.1534 ** (0.0139)	0.1048 ** (0.0126)	-0.8725 ** (0.1456)	-0.3584 ** (0.1248)	0.0063 (0.0524)	-0.0349 (0.0384)
Other variables	Control	Control	Control	Control	Control	Control
Observed value	3627	5626	3627	5626	3627	5626
Part II						
Social Engagementnum	0.0667 ** (0.0078)	0.0510 ** (0.0114)	-0.4670 ** (0.0618)	-0.1343 (0.1029)	-0.0017 (0.0252)	-0.0231 (0.0331)
Other variables	Control	Control	Control	Control	Control	Control
Observed value	3627	5626	3627	5626	3627	5626

Table shows only the second-stage estimation. All models are marginal effects with robust standard errors in parentheses. The marginal values of the first-stage estimation and the control variables are not shown due to length limitations.** *p* < 0.01.

**Table 7 ijerph-16-00278-t007:** How social engagement affects the health of the elderly.

Variable	Intense, Moderately Intense, or Leisure Activities	Outpatient Service
Part I		
Total sample (n = 9353)		
Social Engagement	0.0870 ** (0.0094)	0.0323 ** (0.0089)
Rural sample (n = 5626)		
Social Engagement	0.0896 ** (0.0121)	0.0257 * (0.0115)
Urban sample (n = 3627)		
Social Engagement	0.0871 ** (0.0150)	0.0407 ** (0.0142)
Part II		
Entire sample (n = 9353)		
Social Engagement_num	0.0340 ** (0.0048)	0.0174 * (0.0045)
Rural sample (n = 5626)		
Social Engagement_num	0.0445 ** (0.0071)	0.0163 * (0.0067)
Urban sample (n = 3627)		
Social Engagement_num	0.0263 ** (0.0065)	0.0175 ** (0.0061)

*Note.* Table shows only the regression, with other variables unchanged, which are not reported due to length limitations. All models are marginal effects with robust standard errors in parentheses. * *p* < 0.05, ** *p* < 0.01.

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
