# Peer review of "Social Engagement and Elderly Health in China: Evidence from the China Health and Retirement Longitudinal Survey (CHARLS)"

_ijerph, 2019, doi:10.3390/ijerph16020278_

Round 1

Reviewer 1 Report

The theoretical basis for this study should be more explicitly discussed.There are too many terms used that are not linked together, such as social determinants of health, social capital, and social engagement. Each of these areas has an extensive body of literature and some terms are specifically examined within the context of aging and health. If these were presented within a theoretical framework, it would be clearer how the authors are conceptualizing the interconnections. 

Under "Data," there is a paragraph about the authors and their supporters, which does not seem to fit. Was this misplaced? This section could provide more detail about CHARLS.

The discussion needs to be expanded and tied back to the theoretical framework and hypotheses of the study. 

Author Response

1. The theoretical basis for this study should be more explicitly discussed. There are too many terms used that are not linked together, such as social determinants of health, social capital, and social engagement. Each of these areas has an extensive body of literature and some terms are specifically examined within the context of aging and health. If these were presented within a theoretical framework, it would be clearer how the authors are conceptualizing the interconnections.

Response 1: Thanks for your comments about the concepts and theory framework.

These concepts have been defined by Andersson (1998), Bath & Deeg (2005), Bennett (2002,2005) and Poulsen et al. (2011) so on. More detail has been provided on p.2, lines 54-63 and on p.15, lines 516-525. In our view of point, these definitions are different types of social engagement. For example, while attending church services is an activity or engagement which often has a social capital, it can also stimulate contact with close friends. The overlap is also reflected in studies which have examined two or more of these types of engagement in combinationfor example, Mendes de Leon et al. (2003) [Cf, Mendes De Leon, T. A. Glass, and L. F. Berkman. "Social engagement and disability in a community population of older adults: The New Haven EPESE. " American Journal of Epidemiology 157.7(2003):633-642.] examined the effects of both participations in social activity and social networks on disability.

Based on Pierre Bourdieu’s (1986) social capital theorywe discussed a conceptual model of social engagement processes on elderly health outcomes. The conceptual model is presented on p.4 Figure 2.1.  

2.Under "Data," there is a paragraph about the authors and their supporters, which does not seem to fit. Was this misplaced? This section could provide more detail about CHARLS.

Response 2:  Thank you for pointing out the misplacement! In the revised manuscript, the paragraph about the authors and their supporters has been replaced with more detail about CHARLS, on p.5, lines 173-193.

3.The discussion needs to be expanded and tied back to the theoretical framework and hypotheses of the study.

Response 3: Based on this comment, the discussion has been expanded in the revised manuscript. More detail is provided on p.12, lines 402-413.

Reviewer 2 Report

Review of “Social Engagement and Elderly Health in China: Evidence from the China Health and Retirement Longitudinal Study”

            This paper reports on analysis of data from the CHARLS, data collected in China in 2011 and 2013.  The research question is whether social engagement (measured as a count of activities participated in) is associated with better health (measured as self-rated health, CESD, and chronic conditions), net of some contextual level variables that would make social engagement more possible.  The findings are positive for self-rated health and CESD but not chronic conditions.  There are differences between urban and rural respondents.  Strengths of the manuscript include the large, representative data set and its strong measures of the concepts tested.

1. The paper needs proofreading and editing for English language grammar and usage.

2. Conceptually, the argument is hard to follow, and to differentiate what the authors mean by social engagement and social capital.  On p.2, lines 58-59, “social capital” is said to have three components: Social participation, social networks, and social support.  But on p.3, lines 135-136 it is “social engagement” that reflects the individual’s social network, social participation, and social support.  On p. 3 line 98, social capital is described as an umbrella concept but in the same sentence the pronoun “its” seems to refer to social engagement and the difficulty of establishing “its” aggregate impact.  In general both paragraphs on p. 3 are wandering and unclear.

3. P. 4 introduces the contextual level variables of the availability of activity centers and bus lines, ostensibly to solve what the authors describe as the endogeneity problem.  In the social capital literature with which I am familiar, these community-level variables would be considered of interest on their own, and hierarchical linear models would be employed to account for the two levels, and the nested levels of data.  I am not familiar with the 2SRI method, so perhaps that is doing the same thing, but it seems to be a solution to different issues regarding unobserved variables and reverse causality.    

4. There is no description of the sampling method, its probability basis, stratification, response rate, weighting, oversampling, etc. 

5. The study design is also unclear.  Are these two waves of data with the same respondents?  If so, what is the attrition rate?  From which wave are the variables taken?  This is very unclear.

6. It is unusual to provide the funding details in the middle of the paper.

7. Why is the measure of self-rated health dichotomized?  No justification is given, and a great deal of information is lost if those who classify their health as excellent are grouped with those who say good.

8. On p. 5 the description of the measure of mental health does not mention that this is the CESD scale, thus it was a surprise to read this on the next page.

9. It is odd to justify the inclusion of chronic conditions “to address potential bias related to misreporting” when the chronic conditions are self-reported. 

10. The measure of activities has some problems.  Two of the items (investing in stocks and surfing the Internet) are solitary, not social activities.  “Other activity” is dropped, although it is not clear why it is said to be in conflict with the other items.

11. I am confused about the study design and the language used to interpret the findings.  Repeatedly on p. 8 the dependent variables are said to have “increased” or “decreased”.  Is the analysis capturing change over time between 2011 and 2013?  Variables should be clearly labelled for their year of measurement.  The statement “When social engagement became an essential part of daily life, it met their needs and helped them to gain emotional support and relieve distress, resulted in reducing their CESD and improving mental health” is strongly over-interpretive; it implies a measurement of change over time in first social engagement, and then mental health. 

12. Were the differences shown in Table 4 between rural and urban residents tested for statistical significance?

13. The question about exercise on p.10 does not seem very useful if 10 minutes of leisure activity per week is considered exercise.

14. Other concluding statements on p.10 also seem to greatly overstate the findings: lines 335-336; lines 351-353.

Author Response

1.The paper needs proofreading and editing for English language grammar and usage.

Response 1: We address and check the comment about English language grammar and usage in revised manuscript. What’s more, we hired a professional English editor Sharon Lynn Bear for proofreading.

2. Conceptually, the argument is hard to follow, and to differentiate what the authors mean by social engagement and social capital.  On p.2, lines 58-59, “social capital” is said to have three components: Social participation, social networks, and social support.  But on p.3, lines 135-136 it is “social engagement” that reflects the individual’s social network, social participation, and social support.  On p. 3 line 98, social capital is described as an umbrella concept but in the same sentence the pronoun “its” seems to refer to social engagement and the difficulty of establishing “its” aggregate impact.  In general, both paragraphs on p. 3 are wandering and unclear.

Response 2: Thanks for your comments about the concepts and theory framework. These concepts have been defined by Andersson (1998), Bath & Deeg (2005), Bennett (2002,2005) and Poulsen et al. (2011) so on. More detail has been provided on p.2, lines 54-63 and on p.15, lines 516-525.  In our view of point, these definitions are different types of social engagement. For example, while attending church services is an activity or engagement which often has a social capital, it can also stimulate contact with close friends. The overlap is also reflected in studies which have examined two or more of these types of engagement in combinationfor example, Mendes de Leon et al. (2003) [C. f, Mendes De Leon, T. A. Glass, and L. F. Berkman. "Social engagement and disability in a community population of older adults: The New Haven EPESE. " American Journal of Epidemiology 157. 7(2003): 633-642.] examined the effects of both participations in social activity and social networks on disability.

Based on Pierre Bourdieu’s (1986) social capital theorywe discussed a conceptual model of social engagement processes on elderly health outcomes. The conceptual model is presented on p.4 Figure 2.1. 

3. P. 4 introduces the contextual level variables of the availability of activity centers and bus lines, ostensibly to solve what the authors describe as the endogeneity problem.  In the social capital literature with which I am familiar, these community-level variables would be considered of interest on their own, and hierarchical linear models would be employed to account for the two levels, and the nested levels of data.  I am not familiar with the 2SRI method, so perhaps that is doing the same thing, but it seems to be a solution to different issues regarding unobserved variables and reverse causality.

Response 3: In our work, the model may have endogeneity problems. First, social engagement depends on unobservable preferences (such as the confidence in expectations), which may lead to self-selection or missing variables. Second, there may be a reverse causality between social engagement and health, which means that health status could play a role in older adults’ social engagement. So we turn to the instrumental-variable method with two-stage regression estimation to address these two problems.

Thank you for suggesting that these community-level variables would be considered of interest on their own, and hierarchical linear models would be employed to account for the two levels, and the nested levels of data. This is a good idea, although our present work fail to address it. In social capital literature, neighborhood social capital or community-level social capital is important for elderly health and health behavior. But as most of the elderly in China, especially those in rural China, have permanent address, available scales could be lacking and fail to capture multiple dimensions of community-level social capital. So we indicate this issue on p.13, lines 408-413. In future, if possible, we will expand it.

4. There is no description of the sampling method, its probability basis, stratification, response rate, weighting, oversampling, etc.

 Response 4: The China Health and Retirement Longitudinal Study (CHARLS) has received critical support from Peking University, the National Natural Science Foundation of China, the Behavioral and Social Research Division of the National Institute on Aging and the World Bank. CHARLS is a nationally representative longitudinal survey of persons in China 45years of age or older and their spouses, including assessments of social, economic, and health circumstances of community-residents [ Zhao, Y., Hu, Y., Smith, J. P., Strauss, J., and Yang, G. (2014), “Cohort profile: The China Health and Retirement Longitudinal Study (CHARLS)”, International Journal of Epidemiology, Vol. 43, No.1, pp. 61-68.]. All data will be made public one year after the end of data collection. CHARLS adopts multi-stage stratified PPS sampling. As an innovation of CHARLS, a software package (CHARLS-GIS) is being created to make village sampling frames.

This point is being to expand at revised manuscript on p.5 lines 174-182.

5. The study design is also unclear.  Are these two waves of data with the same respondents?  If so, what is the attrition rate?  From which wave are the variables taken?  This is very unclear.

 Response 5: The CHARLS respondents are followed up every two years, using a face-to-face computer-assisted personal interview. Furthermore, the CHARLS questionnaire includes the following modules: demographics, family structure/transfer, health status and functioning, biomarkers, health care and insurance, work, retirement and pension, income and consumption, assets (individual and household), and community level information.

The baseline national wave of CHARLS is being fielded in 2011 and includes about 10,000 households and 17,500 individuals in 150 counties/districts and 450 villages/resident committees (or villages) from 28 provinces.

This point is being to expand at revised manuscript on p.5 lines 182-193.

6. It is unusual to provide the funding details in the middle of the paper.

 Response 6: In revised manuscript, the funding details has been laid the funding on p.13, lines 437-448; on p.14, lines 449-450.

7. Why is the measure of self-rated health dichotomized?  No justification is given, and a great deal of information is lost if those who classify their health as excellent are grouped with those who say good.

 Response 7: One of the basic principles of category merging is to combine similar categories together and to maximize differences between categories in terms of health and physical function, and there is a significant relationship between depression symptoms. This study calculated the scores of different self-rated health categories in terms of mental health and the number of chronic diseases. More details about self-rated health are provided in the Table 1. It is found that the difference between their health as “excellent” or “very good” and “good” is smaller than that of those health as "good" and "poor" or “bad”, so "excellent", “very good” and "good" are combined.

More detail is provided at the revised manuscript on p.5, lines 202-209 and Table 1.

8. On p. 5 the description of the measure of mental health does not mention that this is the CESD scale, thus it was a surprise to read this on the next page.

 Response 8: It is correct that the description of the measure of mental health does not mention that this is the CESD scale, and we apology to you, our distinguished reviewer. In the revised manuscript, we use continually the degree of psychological distress to measure mental health.

9. It is odd to justify the inclusion of chronic conditions “to address potential bias related to misreporting” when the chronic conditions are self-reported.

 Response 9: On the CHARLS, the all chronic conditions are diagnosed by a doctor rather than self-reported.

10. The measure of activities has some problems.  Two of the items (investing in stocks and surfing the Internet) are solitary, not social activities.  “Other activity” is dropped, although it is not clear why it is said to be in conflict with the other items.

 Response 10: On the CHARLS, the measure of activities are interacted with friend; play Ma-jong, play chess, play cards, or went to community club; provided help to family, friends, or neighbors who do not live with you and did not pay you for the help; went to a sport, social, or other kind of club; took part in a community-related organization; done voluntary or charity work; cared for a sick or disabled adult who does not live with you and who did not pay you for the help; attended an educational or training course; Stock investment; used the Internet; other; none of these. What’s worthy, the interview cannot select” None of these” together with any other answer.  

Two of the items (investing in stocks and surfing the Internet) are excluded because they are not participatory in nature.

11. I am confused about the study design and the language used to interpret the findings.  Repeatedly on p. 8 the dependent variables are said to have “increased” or “decreased”.  Is the analysis capturing change over time between 2011 and 2013?  Variables should be clearly labelled for their year of measurement.  The statement “When social engagement became an essential part of daily life, it met their needs and helped them to gain emotional support and relieve distress, resulted in reducing their CESD and improving mental health” is strongly over-interpretive; it implies a measurement of change over time in first social engagement, and then mental health.

 Response 11: These false have been revised in current manuscript on p.9, lines 277-316 and on p.10, lines 317-318.

12. Were the differences shown in Table 4 between rural and urban residents tested for statistical significance?

 Response 12:  The results show that social engagement significantly influenced the self-rated health and degree of psychological distress of the urban and rural residents, when social engagement has not affected the number of chronic conditions.

13. The question about exercise on p.10 does not seem very useful if 10 minutes of leisure activity per week is considered exercise.

 Response 13: On the CHARLS, respondents are asked to think only about those physical activities that they did for at least 10 minutes at a time. So, to our knowledge, it could be useful rather than other indicators. If possible, we will continue to expand the measure and offer more information in future work.

14. Other concluding statements on also seem to greatly overstate the findings:

Response 14: These statements, which seem to overstate p.10, lines 335-336; lines 351-353. have been revised. More detail is provided on p.11, lines 353-356, on p.12, lines 368-370 in the revised manuscript.

Reviewer 3 Report

1. I am asking for a correction of the way the literature is quoted from alphabetical to the order of quoting

2. Please, quote the following articles at your manuscript:

Cybulski M, Cybulski L, Krajewska-Kulak E, Cwalina U. The level of emotion control, anxiety, and self-efficacy in the elderly in Bialystok, Poland. Clin Interv Aging. 20178;12:305-314.

Cybulski M, Cybulski L, Krajewska-Kulak E, Cwalina U. Illness acceptance, pain perception and expectations for physicians of the elderly in Poland. BMC Geriatr. 2017;17(1):46

3. I am asking you to adapt the list of references to the journal's guidelines.

Author Response

1. I am asking for a correction of the way the literature is quoted from alphabetical to the order of quoting

Response 1: The authors take a correction of the way the literature in revised manuscript, such as on p.14-16, lines 454-483.

2. Please, quote the following articles at your manuscript:

Cybulski M, Cybulski L, Krajewska-Kulak E, Cwalina U. The level of emotion control, anxiety, and self-efficacy in the elderly in Bialystok, Poland. Clin Interv Aging. 20178; 12: 305-314.

Cybulski M, Cybulski L, Krajewska-Kulak E, Cwalina U. Illness acceptance, pain perception and expectations for physicians of the elderly in Poland. BMC Geriatr. 2017;17(1):46

Response 2: The two articles are quoted at revised manuscript. The articleCybulski, M., Cybulski, L., Krajewskakulak, E., and Cwalina, U. (2017), “The level of emotion control, anxiety, and self-efficacy in the elderly in bialystok, Poland”, Clinical Interventions in Aging, Vol.12, pp.305-314.” on p. 3, lines 122-123 and on p.13, lines 418-420.

 On p.13, line 415, the article “Cybulski M, Cybulski L, Krajewska-Kulak E, Cwalina U. Illness acceptance, pain perception and expectations for physicians of the elderly in Poland. BMC Geriatr. 2017; 17(1): 46” is quoted.

3. I am asking you to adapt the list of references to the journal's guidelines.

Response 3:  Thanks for your suggestion, we adapt the list of references to the journal’s guideline. More details could be reviewed in the revised manuscript.

Round 2

Reviewer 2 Report

The authors have responded to my comments, although I disagree with some of their decisions.  The table showing the association between self-rated health, psychological distress, and chronic conditions, for example, clearly shows a linear association across all categories of SRH -- no tests of difference are conducted to bolster their claim that the variable should be dichotomized. 

There is still a considerable problem with English language grammar and phrasing.  For example, the following are not complete sentences:

"Neurodegenerative and neuropsychiatric conditions that can affect older people,such as dementia and reduce their overall quality of life [49]."

"There needs much convincing evidence that how social engagement leads to better health."

Author Response

Cover Letter

Dear peer reviewer,

Thank you for these comments about our manuscript.  Here are our responses to explain point-by-point the details to your comments.

1. The authors have responded to my comments, although I disagree with some of their decisions.  The table showing the association between self-rated health, psychological distress, and chronic conditions, for example, clearly shows a linear association across all categories of SRH -- no tests of difference are conducted to bolster their claim that the variable should be dichotomized.

Response 1: Thanks for your comments about the measure of self-rated health. On one hand, this revised manuscript maintains our procedure.

 On the other hand, as you concern that the measure of self-rated health is dichotomized, a great deal of information could be lost if those who classify their health as excellent are grouped with those who say good. So we supplement the information on the effect of social engagement on different self-rated health status. More detail is provided at the revised manuscript on p.9, lines 296-305 and Table 4.

2.There is still a considerable problem with English language grammar and phrasing.  For example, the following are not complete sentences:

Response 2: We again address and check the comment about English language grammar and usage in revised manuscript. What’s more, besides Sharon Lynn Bear, a professional English editor for proofreading, we also hired a peer Darong Dai who received his Ph.D. degree from Texas A&M University in 2018. 

(1)"Neurodegenerative and neuropsychiatric conditions that can affect older people,such as dementia and reduce their overall quality of life [49]."

Response 2(a): The mistake has been addressed on p.3, lines 121-124.

 (2)"There needs much convincing evidence that how social engagement leads to better health."

Response 2(b): The mistake has been addressed on p.12, lines 377-380.
